# Monetary policy reaction function: A Bayesian analysis for the BRICS

**Farah Waheed**[1]*, **Abdul Rashid**[2], **Asma Basit**[3], **Lubna Maroof**[3]

1 Department of Management Studies, Bahria University, Islamabad, Pakistan, 2 International Institute of Islamic Economics (IIIE), International Islamic University, Islamabad, Pakistan, 3 Department of Business Studies, Bahria University, Islamabad, Pakistan

* fwaheed.buic@bahria.edu.pk

## Abstract

This study estimates the monetary policy reaction function (MPRF) in a Dynamic Stochastic General Equilibrium (DSGE) framework using Bayesian analysis for the emerging economies. DSGE models are suitable for the policy analysis because of their simplicity and prominent role of forward-looking variables. This is a pioneer study investigating the combined effects of credit spreads, fiscal imbalances, and monetary autonomy on interest rates for BRICS member countries. Using real data for the period 1970–2021, the posterior estimates confirm that both credit spread and fiscal imbalance significantly contribute to fluctuations in output, inflation, and interest rates in all the sample economies. The estimates show that fluctuations in the inflation rate are due to supply shocks. The empirical estimates also reveal that fiscal imbalances shock significantly affect output in Brazil, India, and South Africa, whereas, based on real data inflation and interest rate are significantly affected by fiscal imbalance shocks in China and South Africa. Yet, the findings suggest that the effects of various shocks on output and interest rates vary across countries.

**Data Availability Statement:** All relevant data used for analysis in the paper, are publicly available. Data for all macroeconomic variables have been collected from the World Bank (WDI) world development indicators (https://databank.

## 1. Introduction

Monetary policy is an important tool to deal with the economic forces in an economy. It is the major element of national stabilization policy in several countries across the globe [1–3]. The primary purpose of a monetary policy is to improve the economy by stabilizing commodity prices, exchange rates, and interest rates by enhancing savings and investments, and hence economic development and growth. No doubt, the transmission mechanism is the most studied area of the monetary economics [4]. In principle, the central bank generally has to take care of variations in exchange rates and foreign interest rates along with the variability in output and inflation gaps. Central banks use the monetary policy tools including open market operations, required reserve ratio, and the discount rate, to immediately affect the interest rate and money supply in the economy [5]. Assessing the impact of monetary policy on economic activity through policy rules is vital. Benchimol and Fourcans [6] state that objectives of the central bank can be best achieved through monetary policy rules (MPRs). The study documented that for each period, a specific MPRF performs better. Hence, the central banks should

worldbank.org/source/world-development-indicators), IFS (https://data.imf.org/?sk=4c514d48-b6ba-49ed-8ab9-52b0c1a0179b), and FRED (https://fred.stlouisfed.org).

**Funding:** The authors received no specific funding for this work.

**Competing interests:** The authors have declared that no competing interests exist.

regularly refresh their estimates for an effective monetary policy. The BRICS member economies pose unique challenges due to unclear business cycles and diverse central bank responses. These economies often struggle with both excess capacity and high inflation, adding complexity to the analysis.

In the recent era, some policy rules have gained an extensive attention for designing a visible and an effective monetary policy [7]. These rules comprise of theoretical rules that respond to intermediate and final targets, and backward- and forward-looking rules. These rules also sometimes incorporate interest rate smoothening term in the policy reaction function (PRF) [5, 8, 9]. Simple MPR include Taylor and Henderson (1993) and Ball's MCI (Monetary Condition Index)—based rule [10]. Taylor and Henderson's rule was based on the assumption of a closed economy. It is MCI-rule performed poorly when exchange rate shocks were introduced in the rule [1]. Therefore, we need a modified PRF.

Indeed some scholars have modified these rules by incorporating variables like the exchange rate, and financial frictions [1, 11, 12]. Yet, most published studies have not systematically incorporated these factors in the PRF. When foreign borrowings are taxed, it will result in discouraging the borrowers and lenders in the international markets. If capital controls are imposed, then the central authority loses its focus on the interest rate in international markets. Rather, it diverts its focus towards the stability of domestic prices [13]. Bekareva and Meltenisova [14] argue that BRICS countries exhibit significant heterogeneity, making it impractical to implement a unified monetary policy with a singular objective for the entire union. Therefore, it would be worthwhile to study them separately. Further, it is noticed that most of the previous studies are based on a single-equation model and thus, they do not provide reliable and robust evidence for establishing the link between the conduct of MP and the economy's performance [2, 15]. Such types of models are now considered far away from the real world. Therefore, there is a dire need to modify and empirically estimate the PRF by incorporating key variables such as credit spread, fiscal imbalance, and monetary autonomy.

In principle, fiscal imbalances, credit spread, and monetary autonomy either directly or indirectly affect the rate of interest. Benchimol, Saadon and Segev [16] argued that before any monetary policy decision uncertainty in the financial market should be monitored. In general, frictions in money and capital markets, frictions to investment flows, and informational frictions are most common frictions that are considered in policy rules. Credit spread is a type of informational friction, which impacts the interest rates [17, 18]. The existing literature concluded that the Taylor rule with an adjusted credit spread performs in a better manner in comparison to without the credit-spread rule [2, 18]. Therefore, for comprehensive understanding of the monetary policy process, this type of informational friction should be part of the PRF to get robust evidence on the interest-rate setting mechanism.

Similarly, there are few researchers who considered fiscal imbalance while examining the interest-rate setting behavior of central banks [19–22]. Yet, it is expected that fiscal imbalances play an important role. Especially, in emerging countries, this role can be more profound as these countries are more likely to indulge in monetary issues. Theoretically, fiscal imbalances have a significant role in the determination of interest rates in an economy. The government finances its deficits through borrowings from the domestic markets, causing private sector to face the scarcity of available funds. Excessive government borrowing not only increases the market rate of interest but also brings the crowding-out effect in the economy [23].

Some prior studies documented that there is a positive correlation between the rate of interest and fiscal imbalances [24–26], while some studies did not agree with this viewpoint [27, 28]. The studies documented that an increase in the budget deficit do not increase the rate of interest because the government has to pay off the outstanding debts in future by imposing higher taxes. Therefore, by anticipating taxes, households reduces their current consumption

and thus, aggregate demand and the rate of interest, remains unaffected. Having such inconclusive arguments, the effect of fiscal imbalances on interest-rate should be investigated.

States are less autonomous because monetary independence is costly [29]. The capital flows are very likely to be in and out of the markets owing to various macro factors [30, 31]. The likelihood of demise of banking industry, currency crisis, and rising inflation are some of the usual outcomes of excessive and sudden outflows of foreign capital. However, foreign capital flows may be controlled in the emerging economies. This is important because abrupt and substantial flows of foreign capital are expected to result in financial instability and currency crisis in the economy. According to imminent economists, the extent of central authority's autonomy impacts money and credit expansions along with large effects on macroeconomic variables e.g. interest rates, budget deficits and the inflation [32–34]. In this context, an analysis of the said phenomenon is the need of the hour, specifically to answer how monetary autonomy plays its role in the interest-rate setting behavior of the emerging economies.

Keeping in view the existing gaps in the empirical literature, this paper empirically estimates the MPRF proposed by Waheed and Rashid [35] for BRICS economies. To the best of authors' knowledge, this study is adding an empirical contribution to the literature by estimating MPRF for the BRICS member countries in a Bayesian framework. Specifically, using annual data for the period 1970–2021, the stochastic behavior of the model is driven by six exogenous disturbances, namely, demand shocks, supply shocks, foreign interest rate shocks, monetary policy shocks, credit spread shocks, and fiscal imbalance shocks. Furthermore, this is a pioneer study investigating the combined effects of credit spreads, fiscal imbalances and monetary autonomy on the interest-rate setting mechanism of the emerging economies. The observable variables are output, inflation, the interest rate, fiscal imbalances, credit spread, and the foreign interest rate. USA interest rate is considered as a proxy for the foreign interest rate in our empirical analysis.

The posterior estimates confirm that both credit spread and fiscal imbalances significantly contribute to fluctuations in inflation rates, output level, and the interest rate. The supply shock significantly affects output, interest rate and inflation in all the sample economies. Except Russia, the credit spread shock significantly affects inflation in all the sample emerging economies. The empirical estimates show that fiscal imbalance shocks significantly affect output in Brazil, India, and South Africa, whereas, inflation and interest rates are significantly affected by fiscal imbalance shocks in China and South Africa.

The rest of the paper proceeds as follows. Monetary policy rules are presented in Section 2. The model is presented in Section 3. Data and choice of priors are discussed in Section 4. Section 5 comprises of empirical analysis. The concluding remarks are given in Section 6.

## 2. Monetary policy rules

The concept of MPR is eye-catching for many reasons, especially due to accountability and transparency of the central authority. Policy rules gather the knowledge about the effective operation of the policy and can also transfer this type of knowledge to future generations. As far as rules of monetary policy are concerned, theoretically, it has been divided into two kinds; targeting rules and instrument rules.

### 2.1 Targeting rules

This approach was advocated by Svensson [36]. It better captures monetary policy in the output and inflation targeting countries. Under targeting rules, the behavior of central authority is defined according to the objective function of a central bank that follows particular targets. The targeting rules are further categorized as general targeting rules and specific targeting

rules. In general targeting, the central bank has an operational loss function and the objective of the monetary policy is the minimization of the loss function.

As far as the specific targeting is concerned, the central bank collects data and then formulates the policy accordingly in a usual complex way. The particular conditions for setting instruments are specified in the specific targeting rules. The targeting rules include the target variable such as an inflation target. However, there can be additional targets e.g. the exchange rate targeting. Hence, the policy can be labeled as "flexible inflation targeting" as per Svensson [37].

In the 1990s, various central banks have chosen the structured framework to put the inflation rate and output level on the right path. The framework adopted by central banks lacked prior research footing [38]. Later on, empirical research in monetary policy were carried out which gave rise to the structure known as "inflation forecast targeting" [1, 16, 39]. The designed framework was akin to the rule where discretion was given to central banks in the stipulated limits. Such regulations were labeled as targeting rules [36]. In such frameworks, the central bank proclaims the aims of inflation target and monetary policy. This process is characterized by the role of monetary policy transparency and accountability of concerned authorities. Inflation projection is considered as the intermediary target in this framework. To keep forecasted inflation on target, the central bank sets various instruments. Bernanke and Mishkin [40] stated this framework as "constrained discretion". Taylor [5] worked on the transmission mechanism and different channels of monetary policy transmission were also discussed. His results proof that countries with a clear inflation target have achieved lower inflation rates as compared to countries without the target. On the contrary, output in the inflation-targeting countries is more volatile than non-inflation targeting countries. Nikolsko-Rzhevskyy, Papell [41] documented that rules versus discretion debate has evolved to "policy rules versus constrained discretion". They stated that economic performance is much better in low deviation periods. Similarly, the estimates show that rules with larger coefficients on the inflation gap are preferred over the other rules. The study also directed that this kind of rule should be added to the semi-annual MP report of the Fed. Benchimol and Ivashchenko [42] documented that the nonlinearities can occur as the economy is affected by the domestic and foreign financial markets. The study documented that the policymakers should use Switching volatility shocks (SVSs) models; if they use standard linear models they might overlook non-linear dynamics and interaction among financial markets and the economy.

Friedman [43] also discussed MPRs. Friedman and Taylor are at two different poles as far as MPRs are concerned. Friedman advocated a constant money growth rule, whereas Taylor's emphasis was on interest rate rules. According to Friedman monetary policy impacts the inflation with a lag. Therefore, current inflation is not suitable for targeting. Svensson [36] stated that the central bank which has not any information except current inflation and the output gap would adopt the "Taylor rule". In contrast, if the central bank can easily access data of monetary aggregates, then it would go for Friedman's rule of k-percent money growth. According to this rule, the money supply should be increased at a constant rate every year without focusing on the state of the business cycle.

## 2.2 Instrument rules

Instrument rules imply that an instrument of monetary policy helps regulate the function of the state of the economy. Instrument rules are easy to follow and they do not require much technical information. These rules are logical in the sense that agreement to these rules can easily be verified. They do not require many professionals. Rather, only few professionals can easily accomplish this sort of task. They just take decisions according to the prevalent

conditions of the economy. The only requirement is to adopt a rule having theoretical properties and tailored to perform well. The implementation of monetary policy comprises of numerous rules and practices. These are commonly known as "operating procedures". The objective of examining these operating procedures is to know the instruments which are under the control of central authority, the factors responsible for the choice of instruments, and also the way these instruments affect the interest rate [44].

The monetary authority has to choose between an interest rate and monetary aggregate as a policy instrument. This analysis was done by Poole [45]. According to Poole, information on the market rate is available continuously, while output and inflation are not immediately available. They are available on a monthly or quarterly basis. The authority lacks information if it plans to set policy by observing the disturbances in goods and money markets. Therefore, the identification of the exact nature of disturbances is not possible from just the movement in the market rate of interest. Poole's stated that through comparison of the variances of output for both the alternatives, policymakers can decide to keep either market rate of interest or market quantity constant. The one with the lower variances will be preferred over the other.

Expectations, inflation, and supply disturbances are ignored in the Poole's model. These factors are taken into consideration in other models [46]. Friedman [47] also criticized the Poole's model. He stated that an objective function is not simply the variances of output when inflation is included in the model. According to Friedman, the instrument choice is an endogenous decision of policymaker and it is dependent on the objectives of the monetary policy.

Forward- and backward-looking MPRF was derived by Rashid and Waheed [9]. By using the IV-GMM approach, the estimates of Pakistan showed that both forward-and backward-looking variables should be part of the PRF as it helps the policymakers to set the interest rate accurately. Both lags and leads of these variables (output, inflation, and the exchange rate) are found to be significant in determining the rate of interest.

Taylor [48] developed an instrument rule for the operation of monetary policy well known in the literature as "Taylor rule". The purpose of designing such type of rule was to provide information that how a central bank sets its short-term interest rate to achieve its desired objectives, for example; output stabilization in the economy and also to keep inflation under control. One version of Taylor rule is given as:

$$i_t = r^* + \pi^* + \beta(\pi_t - \pi^*) + \gamma(y_t - y_N) \tag{1}$$

where,
   $r^*$ = Average long-run real interest rate
   $\pi^*$ = Targeted inflation rate
   $\pi_t$ = Current inflation rate
   $y_t$ = Current output

It is critical to mention here that Taylor [48] clarified avoiding the mechanical use of the rule rather the rule should be taken as a guideline while making decisions about monetary policy. For policymakers, it is recommended not to deviate from the rule [36]. The rule guides the central authority in decision making. Such simplicity facilitates the entire process and helps in avoiding problems. An outside observer can easily determine the type of rule followed by the central bank. However, it becomes difficult for the decision-making authority to move away from the declared rule. It can be concluded that the instrument rules are simple, reliable but strict. However, commitment to these rules is technically feasible.

## 3. Monetary policy reaction function

Our model comprises of three major sectors viz. household, intermediary sector, which is the central bank of the country, and corporate firms. The analytical framework for this study is presented in this section. The detailed derivation of the model is available in supporting information (*S.1 Derivation of Monetary Policy Reaction Function in S1 Appendix*). The equations of the model are as follows:

$$\widehat{Y}_t = -\bar{\sigma}(\widehat{i}_t^{avg} - E_t\pi_{t+1}) + E_t\widehat{Y}_{t+1} - E_t\Delta\widehat{\zeta}_{t+1} - E_t\Delta\widehat{e}_{t+1} - \bar{\sigma}s_\Omega\widehat{\Omega} + \bar{\sigma}(s_\Omega + \psi_\Omega)E_t\Omega_{t+1} + \varepsilon_y \quad (2)$$

where,

$$\widehat{i}_t^{avg} = \pi_b\widehat{i}_t^b + \pi_s\widehat{i}_t^d \qquad\qquad \Psi_\Omega = \pi_b(1 - \chi_b) - \pi_s(1 - \chi_s)$$

$$e_t = s_c\bar{c}_t + \widehat{E}_t$$

The inflation rate is determined as:

$$\pi_t = \xi(\omega_y\widehat{Y}_t - \widehat{\bar{\lambda}}_t - \nu\bar{h}_t - (1 + \omega_y)\widehat{Z}_t + \widehat{\mu}_t^w + \widehat{\tau}_t) + \beta E_t\pi_{t+1} + \varepsilon_\pi \quad (3)$$

where, $\widehat{\bar{\lambda}} \equiv \log\left(\frac{\widehat{\lambda}_t}{\bar{\lambda}}\right), \bar{h}_t \equiv \log\left(\frac{H_t}{H}\right), \widehat{Z}_t \equiv \log\left(\frac{Z_t}{Z}\right), \widehat{\mu}_t^w \equiv \log\left(\frac{\mu_t^w}{\bar{\mu}^w}\right),$

$$\widehat{\tau}_t \equiv -\log\frac{1 - \tau_t}{1 - \bar{\tau}}, \text{ and } \xi \equiv \frac{1 - \alpha}{\alpha} - \frac{1 - \alpha\beta}{1 + \omega_y\theta} > 0$$

The MPRF is:

$$\widehat{i}_t^{avg} = \bar{i} + \theta_p\pi_t + \theta_y\widehat{Y}_t + \theta_f i_t^* + \theta_n\widehat{\zeta}_t + \theta_\omega\widehat{\omega}_t + \widehat{m}_t + \varepsilon_i \quad (4)$$

Where, $\theta_f = \theta_f^c + \theta_f^o$ represent close and open capital accounts, respectively.

Eqs (2), (3), and (4) represent complete system of equations of the model. Eq (2) shows the law of motion for private indebtedness. It shows that central bank can encourage or discourage capital inflow through the instrument of capital tax or subsidy, respectively. Eq (3) is the inflation determination equation. Eq (4) is the interest rate determination equation. $\theta_f$ is normalized on 0–1 scale, 0 represents completely closed capital account and 1 represent completely open capital account. When capital account is open, the interaction term i.e., $K_{it}$ representing capital account openness index in the particular country will be added to the PRF.

## 4. Estimation technique and choice of priors

### 4.1 Estimation technique

Dynamic Stochastic General Equilibrium (DSGE) models have attained attention of academic researchers and are widely used by several policy making-institutions. Various central banks have sought to estimate their DSGE models like Bank of England, European Central Bank, Federal Reserve Board, and Bank of Sweden. Economies have started to employ such models for the purpose of forecasting [49]. Following Wong and Eng [50], An and Schorfheide [51], Smets and Wouters [52], and [49], Adolfson, Laséen [53], we applied the Bayesian approach for the empirical estimation of the derived rule. The incorporation of prior information into the estimation process and model misspecification dealing are some of the key properties of Bayesian technique. This is of particular importance, especially, when small data samples are available [52]. The Bayesian method is appropriately defined [54]. When a set of observables

**Table 1. Parameters selection for the Bayesian estimation.**

| Symbol | Value | Reference |
|---|---|---|
| $\alpha$ | 0.66 | Kolasa and Lombardo [58] |
| $\beta$ | 0.987 | Davis and Presno [59], Cúrdia and Woodford [18], Heathcote and Perri [60], Smets and Wouters [52]. |
| $\kappa$ | 0.0589 | Cúrdia and Woodford [18] |
| $\bar{\sigma}$ | 6.24864 | Cúrdia and Woodford [18] |
| $\omega_y$ | 0.473 | Cúrdia and Woodford [18] |
| $\gamma_b$ | 0.105 | Cúrdia and Woodford [18] |
| $B_\lambda$ | -0.6262 | Cúrdia and Woodford [18] |
| $\omega_X$ | 0.980 | Cúrdia and Woodford [18] |
| $\omega_E$ | 0.306 | Cúrdia and Woodford [18] |

$Y^T$ for a time period T and set of priors $p(\theta)$ are given, then the posterior density of the parameter ($\theta$) is defined as:

$$p(\theta|Y^t) = \frac{\mathcal{L}(\theta|Y^t)p(\theta)}{\int \mathcal{L}(\theta|Y^t)p(\theta)d(\theta)}$$

Certain parameters used in the model of the current study are adopted from the extant literature; these are reported in Table 1. The necessary condition is that the number of endogenous variables should be equal to number of equations. The identification of the parameters is done according to the Jacobian of the steady-state, reduced form matrices, and the first two moments [55–57].

For the process of shocks, we follow a first-order auto-regressive process with an *i.i.d* error term [13, 18, 53, 61]. Specifically, we define shock as:

$$\varepsilon_t^i = \rho^i \varepsilon_{t-1}^i + e_t^i \tag{5}$$

where, $i = D,S,m,f,\omega,$ *and* $\zeta$ denote demand, supply, monetary policy, foreign interest rate, credit spread, and fiscal imbalances shocks, respectively. The coefficient of autocorrelation ($\rho$) undertakes various values for multiple shocks. For demand, supply, fiscal imbalances, and credit spread shocks, the coefficient takes the value of 0.9. However, for monetary policy and foreign interest rate shocks, autocorrelation coefficient takes the value of 0.75 and 0.59, respectively.

## 4.2 Choice of priors

With the information from the data and prior distributions, we apply a Markov Chain Monte Carlo (MCMC) method for the estimation. We choose the priors from the literature as well as stylized facts in a particular country. The prior distributions are assumed as independent across different parameters. By maximizing the log posterior function, we estimate the mean of the posterior distribution, which is the product of prior information of the parameters and the likelihood of the data. The next step is to evaluate the marginal likelihood of the model and to have a complete picture of the posterior distribution; Metropolis-Hastings algorithm is used.

We use beta distribution for the parameters taking values between 0 and 1. For the exogenous shock process coefficient i.e. demand, supply, monetary policy, foreign interest rate, fiscal imbalances, and credit spread shocks follow the beta distribution. Similarly, the gamma distribution is considered for the parameters whose value exceeds 1. For the unbounded parameters,

we assign the normal distribution. The inverse gamma distribution is followed by the standard deviations of the exogenous processes, as it is commonly used in DSGE models for the exogenous shocks [16, 62].

## 5. Empirical analysis

### 5.1 Posterior parameter estimates

We apply the Bayesian approach by using real data for the BRICS economies. Dynare tool in MATLAB is used for the estimation. The univariate and multivariate convergence diagnostics were applied. Both within sequence value and the sum of within sequence statistics and a between sequence variance graphs converge and settle down with time.

S1 Fig (*sub-figures A1.1-A1.5 in S1 Appendix)* shows prior and posterior distribution of the structural parameters clearly indicates that the estimated posteriors are different from the priors, indicating adequate information is driven from the data.

When we normalized the financial openness indicator, which is measured by Chinn-Ito Index on 0–1 scale, on average these countries take the value of 0. Hence, the interaction term "K" in the foreign interest rate term is not there. In the last two columns of Table 2, the posterior mean and confidence interval of all parameters are reported for each country. The confidence intervals at the 5% and 95% percentiles are reported in the square brackets.

In Brazil, the foreign interest rate coefficient ($\theta_f$), output coefficient ($\theta_y$), the inflation coefficient ($\theta_p$), fiscal imbalances coefficient ($\theta_\zeta$), credit spread coefficient ($\theta_\omega$), and interest rate smoothening parameter ($\bar{i}$) are significant. The posterior means are close enough to the prior means except for output and inflation parameters. These priors and posterior distributions for Brazil are graphically presented in the appendix. For the exogenous processes, all the shocks are persistent. The confidence interval confirms that all these coefficients are significant. The log data density (Laplace approximation) is reported to be -223.609. The acceptance ratio for Brazil is 49% and 47%, respectively. The findings are consistent with the findings of De Castro, Gouvea [63], and Silveira [64].

For Russia, the estimates show that the coefficients are significant. The output coefficient ($\theta_y$) is very high as compared to other countries. It shows that the Russian economy is more concerned about output. It is also far from the prior mean. The confidence interval confirms the significance of this variable. The inflation coefficient ($\theta_p$), fiscal imbalances ($\theta_\zeta$), and credit spread ($\theta_\omega$) coefficients are close to their respective priors and all these are significant. The shock processes are also significant, confirming persistence. However, the standard deviation for credit spread shock is insignificant. The log data density (Laplace approximation) is -387.118 for Russia. The acceptance ratios are 43% and 42%, respectively. The findings are consistent with some of the prior studies [26, 65, 66].

In case of India, the estimates show that the coefficients of foreign interest rate ($\theta_f$), output ($\theta_y$), inflation ($\theta_p$), fiscal imbalances ($\theta_\zeta$), credit spread ($\theta_\omega$) and interest rate smoothening parameter ($\bar{i}$) are significant. Except output and inflation coefficient, other posterior estimates are close to their respective priors. The interest rate smoothing parameter is high in India as compared to other emerging countries, which implies that India has the willingness for smoothing the movement of the interest rate. For the shock processes, the estimates indicate these are significant. However, the standard deviation for supply and fiscal imbalances shocks are insignificant as the confidence interval is wider for these shocks. The log data density (Laplace approximation) for India is reported to be -422.387. The acceptance ratios are 41% and 42%, respectively. The findings of the study are coherent with the previous findings of Levine, Vasco, and Yang [67] and Banerjee and Basu [68].

**Table 2. Priors and posteriors distributions–Structural parameters.**

| Parameter | Prior | | | Posterior | |
|---|---|---|---|---|---|
| | Density | Mean | SD | Mean | 90% Interval |
| **Brazil** | | | | | |
| $\theta_f$ | B | 0.50 | 0.20 | 0.4968 | [0.2368, 0.7744] |
| $\theta_y$ | B | 0.50 | 0.25 | 1.3980 | [1.1523,1.6787] |
| $\theta_p$ | B | 1.50 | 0.75 | 0.8482 | [0.8209 0.8757] |
| $\theta_\zeta$ | B | 0.50 | 0.20 | 0.4841 | [0.2069, 0.7635] |
| $\theta_\omega$ | B | 0.50 | 0.20 | 0.4388 | [0.3171, 0.6032] |
| $\bar{\bar{i}}$ | B | 0.50 | 0.20 | 0.4262 | [0.1674, 0.7212] |
| Log Data Density -223.609 | | | | | |
| **Russia** | | | | | |
| $\theta_f$ | B | 0.50 | 0.20 | 0.6469 | [0.3924, 0.9541] |
| $\theta_y$ | B | 0.50 | 0.25 | 1.5957 | [1.3033, 1.9550] |
| $\theta_p$ | B | 1.50 | 0.75 | 0.8363 | [0.7538, 0.9129] |
| $\theta_\zeta$ | B | 0.50 | 0.20 | 0.4655 | [0.2124, 0.8007] |
| $\theta_\omega$ | B | 0.50 | 0.20 | 0.5534 | [0.3102, 0.8164] |
| $\bar{\bar{i}}$ | B | 0.50 | 0.20 | 0.5301 | [0.2680, 0.8022] |
| Log Data Density -387.118 | | | | | |
| **India** | | | | | |
| $\theta_f$ | B | 0.50 | 0.20 | 0.7239 | [0.3658, 0.9968] |
| $\theta_y$ | B | 0.50 | 0.25 | 1.4469 | [1.2367, 1.6956] |
| $\theta_p$ | B | 1.50 | 0.75 | 1.0947 | [1.0484, 1.1427] |
| $\theta_\zeta$ | B | 0.50 | 0.20 | 0.5330 | [0.2204, 0.8629] |
| $\theta_\omega$ | B | 0.50 | 0.20 | 0.4661 | [0.1360, 0.7170] |
| $\bar{\bar{i}}$ | B | 0.50 | 0.20 | 0.8668 | [0.7607, 0.9863] |
| Log Data Density -422.387 | | | | | |
| **China** | | | | | |
| $\theta_f$ | B | 0.50 | 0.20 | 0.4334 | [0.1101, 0.7456] |
| $\theta_y$ | B | 0.50 | 0.25 | 0.0158 | [0.0078, 0.0234] |
| $\theta_p$ | B | 1.50 | 0.75 | 1.6604 | [1.5945, 1.7219] |
| $\theta_\zeta$ | B | 0.50 | 0.20 | 0.5125 | [0.1841, 0.8359] |
| $\theta_\omega$ | B | 0.50 | 0.20 | 0.5258 | [0.1934, 0.7914] |
| $\bar{\bar{i}}$ | B | 0.50 | 0.20 | 0.4515 | [0.1524, 0.7693] |
| Log Data Density -414.473 | | | | | |
| **South Africa** | | | | | |
| $\theta_f$ | B | 0.50 | 0.20 | 0.5092 | [0.3495, 0.6465] |
| $\theta_y$ | B | 0.50 | 0.25 | 1.4909 | [1.2980, 1.7409] |
| $\theta_p$ | B | 1.50 | 0.75 | 1.3803 | [1.2242, 1.5210] |
| $\theta_\zeta$ | B | 0.50 | 0.20 | 0.7254 | [0.559, 0.9467] |
| $\theta_\omega$ | B | 0.50 | 0.20 | 0.6929 | [0.5574, 0.8303] |
| $\bar{\bar{i}}$ | B | 0.50 | 0.20 | 0.7475 | [0.5704, 0.9035] |
| Log Data Density -462.041 | | | | | |

**Note:** Abbreviations used for the priors distributions are: B: Beta; N: Normal; G: Gamma; IG: Inverse Gamma.

The estimates for China show that the parameters of foreign interest rate ($\theta_f$), output ($\theta_y$), credit spread ($\theta_\omega$), fiscal imbalances ($\theta_\zeta$), and inflation ($\theta_p$), are close to their respective priors. The confidence interval shows that all these estimates are statistically significant. The interest

rate smoothing parameter $(\bar{i})$ is also significant. All the shock processes are significant in China, showing persistency in the shocks. However, the standard deviation for monetary policy shock is insignificant. The log data density for the China is -414.473. The acceptance ratios are 45% and 46%, respectively. The estimates are consistent with Ma [69] and Zheng and Guo [62].

The estimates for South Africa show that the parameters of foreign interest rate ($\theta_f$), output ($\theta_y$), credit spread ($\theta_\omega$), fiscal imbalances ($\theta_\zeta$), and inflation ($\theta_p$), are close to their respective priors. The confidence interval shows that these estimated values all these are significant. The interest rate smoothening parameter $(\bar{i})$ is also significant. The posterior mean of $(\bar{i})$ is high showing the evidence of partial adjustment. It also indicates that South Africa has a strong willingness to smooth the interest rate movement. All the shock processes are significant in South Africa, showing persistency in shocks. The standard deviation for demand and monetary policy shocks are insignificant. The log data density (Laplace approximation) is reported to be -462.041. The acceptance ratios are 39% and 43%, respectively. The estimates are consistent with the findings of Dagar and Malik [70], Alpanda, Kotzé [71], and Steinbach, Mathuloe [72].

As a whole, the result shows that the data are informative regarding the posterior distribution. The sample countries respond in a similar way with slight differences in the coefficients. The coefficient of the foreign interest rate is reported to be highest in India, while it is lowest in China. Both output and inflation coefficients are higher in the emerging economies. Except for China, in all the sample emerging countries, the output coefficient is higher than the inflation coefficient. The estimates indicate that the sample emerging economies show more concern for output as compared to inflation. However, the inflation coefficient is also significant in these countries [62, 63].

The fiscal imbalances coefficient is highest in South Africa. However, the estimates show that it is lowest in Russia. Fiscal imbalance is considered as the main driving force to inflation in the emerging economies. The estimates also show that the inflation coefficient is highest in South Africa revealing that higher fiscal imbalances lead to higher inflation [25, 26, 73]. Similarly, the credit-spread coefficient is highest in the case of South Africa. The interest rate smoothing parameter $\bar{i}$ is higher in India, and South Africa, showing the evidence of the partial adjustment and also the strong willingness of the central authority to smooth the movement of interest rate in these countries. The posterior probability interval for all the structural parameters for all the countries also confirms the level of significance.

Tables 3 and 4 report the priors and posterior mean and confidence interval for exogenous processes. We have assumed that the shock process follows a first-order auto-regressive process with an *i.i.d* normal error term. The posterior probability interval for all the parameters for all the countries, in this case, confirms the significance level. However, the standard deviation of exogenous processes i.e. $e_D$ for South Africa, $e_\omega$ for Brazil and Russia, and $e_S$ and $e_m$ for Russia have wide probability intervals to make sure the level of significance.

As far as the standard deviation of the exogenous processes are concerned, the wider confidence interval indicating insignificant results for the demand shock. However, to demand shock, most of the emerging economies show significant estimates except South Africa. Except Russia all other have strong and higher level of significance to the supply shock. In the same way, except Russia and South Africa the estimates of all other economies show exogenous processes to be significant.

For the fitness and stability of structural parameters and shocks, we used Kolmogorov-Smirnov test. In case of Brazil and Russia, 78.4% of the prior support gives unique saddle path solution. About 21.6% of the prior support gives indeterminacy. The Smirnov statistics reported inflation parameter having d-stat value of 0.934 and p-value of 0.000 in deriving indeterminacy. In the case of India, 86.6% of the prior support gives unique saddle path solution. In China and South Africa, all the parameter values in the specified ranges give unique saddle

**Table 3. Priors and posteriors distributions–Exogenous processes.**

| Parameter | Prior | | | Posterior | |
|---|---|---|---|---|---|
| | Density | Mean | SD | Mean | 90% Interval |
| Exogenous processes–AR (1) coefficients | | | | | |
| **Brazil** | | | | | |
| $\rho_D$ | B | 0.50 | 0.20 | 0.5496 | [0.2633, 0.8266] |
| $\rho_S$ | B | 0.50 | 0.20 | 0.8429 | [0.8118, 0.8709] |
| $\rho_m$ | B | 0.50 | 0.20 | 0.5454 | [0.2765, 0.8343] |
| $\rho_f$ | B | 0.50 | 0.20 | 0.5997 | [0.3370, 0.8558] |
| $\rho_\omega$ | B | 0.50 | 0.20 | 0.5747 | [0.5622 0.5938] |
| $\rho_\zeta$ | B | 0.50 | 0.20 | 0.5901 | [0.3347, 0.8665] |
| **Russia** | | | | | |
| $\rho_D$ | B | 0.50 | 0.20 | 0.4934 | [0.1673, 0.7661] |
| $\rho_S$ | B | 0.50 | 0.20 | 0.8520 | [0.7738, 0.9266] |
| $\rho_m$ | B | 0.50 | 0.20 | 0.3573 | [0.1192, 0.5989] |
| $\rho_f$ | B | 0.50 | 0.20 | 0.4802 | [0.1632, 0.7463] |
| $\rho_\omega$ | B | 0.50 | 0.20 | 0.3173 | [0.1802, 0.4661] |
| $\rho_\zeta$ | B | 0.50 | 0.20 | 0.5035 | [0.2302, 0.8366] |
| **India** | | | | | |
| $\rho_D$ | B | 0.500 | 0.20 | 0.7298 | [0.5783, 0.8636] |
| $\rho_S$ | B | 0.500 | 0.20 | 0.1291 | [0.1073, 0.1517] |
| $\rho_m$ | B | 0.500 | 0.20 | 0.7329 | [0.5706, 0.9090] |
| $\rho_f$ | B | 0.500 | 0.20 | 0.7312 | [0.5686, 0.8994] |
| $\rho_\omega$ | B | 0.500 | 0.20 | 0.7367 | [0.6872, 0.7794] |
| $\rho_\zeta$ | B | 0.500 | 0.20 | 0.4201 | [0.2673, 0.5411] |
| **China** | | | | | |
| $\rho_D$ | B | 0.500 | 0.20 | 0.8300 | [0.7015, 0.9677] |
| $\rho_S$ | B | 0.500 | 0.20 | 0.7575 | [0.5707, 0.9005] |
| $\rho_m$ | B | 0.500 | 0.20 | 0.3547 | [0.2508, 0.4505] |
| $\rho_f$ | B | 0.500 | 0.20 | 0.7971 | [0.6539, 0.9740] |
| $\rho_\omega$ | B | 0.500 | 0.20 | 0.5850 | [0.4768, 0.7242] |
| $\rho_\zeta$ | B | 0.500 | 0.20 | 0.9483 | [0.9072, 0.9857] |
| **South Africa** | | | | | |
| $\rho_D$ | B | 0.500 | 0.20 | 0.7487 | [0.6729, 0.8578] |
| $\rho_S$ | B | 0.500 | 0.20 | 0.7789 | [0.7202, 0.8503] |
| $\rho_m$ | B | 0.500 | 0.20 | 0.8782 | [0.7791, 0.9719] |
| $\rho_f$ | B | 0.500 | 0.20 | 0.8234 | [0.6695, 0.9626] |
| $\rho_\omega$ | B | 0.500 | 0.20 | 0.6424 | [0.6035, 0.6975] |
| $\rho_\zeta$ | B | 0.500 | 0.20 | 0.8774 | [0.7308, 0.9975] |

**Note:** Abbreviation used for the priors distributions is: B: Beta

path solution and match prior moment restrictions. The Root-Mean-Squared Error (RMSE) of the model are 0.83 (Brazil), 0.76 (Russia), 0.69 (India), 0.67(China) and 0.78 (South Africa).

## 5.2 Variance decomposition analysis for the emerging economies

To capture the impact of various shocks, we present the estimates of variance decomposition for each sample country. Table 5 presents the variance decomposition for output, inflation, and the interest rate for each country. The estimates show that supply shock significantly

**Table 4. Priors and posteriors distributions–Exogenous processes.**

| Parameter | Prior | | | Posterior | |
|---|---|---|---|---|---|
| | Distribution | Mean | SD | Mean | 90% Interval |
| Exogenous processes–standard deviations | | | | | |
| **Brazil** | | | | | |
| $e_D$ | IG | 1.00 | 2.00 | 0.5459 | [0.3368, 0.7979] |
| $e_S$ | IG | 1.00 | 2.00 | 2.1179 | [1.2853, 2.9604] |
| $e_m$ | IG | 1.00 | 2.00 | 0.2648 | [0.1927, 0.3305] |
| $e_f$ | IG | 1.00 | 2.00 | 0.2574 | [0.1958, 0.3145] |
| $e_\omega$ | IG | 1.00 | 2.00 | 18.7037 | [13.9959, 24.3022] |
| $e_\zeta$ | IG | 1.00 | 2.00 | 0.2683 | [0.1978, 0.3518] |
| **Russia** | | | | | |
| $e_D$ | IG | 1.00 | 2.00 | 0.6287 | [0.3061, 0.9567] |
| $e_S$ | IG | 1.00 | 2.00 | 18.5567 | [8.0050, 27.6328] |
| $e_m$ | IG | 1.00 | 2.00 | 13.6743 | [10.5108, 16.2990] |
| $e_f$ | IG | 1.00 | 2.00 | 0.8057 | [0.3371, 1.5459] |
| $e_\omega$ | IG | 1.00 | 2.00 | 18.0318 | [11.4800, 28.5940] |
| $e_\zeta$ | IG | 1.00 | 2.00 | 1.2975 | [0.3604, 2.2109] |
| **India** | | | | | |
| $e_D$ | IG | 1.000 | 2.00 | 0.5584 | [0.2665, 0.7962] |
| $e_S$ | IG | 1.000 | 2.00 | 4.7303 | [3.9985, 5.4175] |
| $e_m$ | IG | 1.000 | 2.00 | 0.7080 | [0.3361, 1.1459] |
| $e_f$ | IG | 1.000 | 2.00 | 0.8587 | [0.3505, 1.3863] |
| $e_\omega$ | IG | 1.000 | 2.00 | 0.3734 | [0.2289, 0.5049] |
| $e_\zeta$ | IG | 1.000 | 2.00 | 5.4955 | [4.5536, 6.4131] |
| **China** | | | | | |
| $e_D$ | IG | 1.000 | 2.00 | 0.6216 | [0.2964, 1.0104] |
| $e_S$ | IG | 1.000 | 2.00 | 1.8786 | [1.5144, 2.2891] |
| $e_m$ | IG | 1.000 | 2.00 | 5.3996 | [4.2317, 6.4567] |
| $e_f$ | IG | 1.000 | 2.00 | 0.6842 | [0.3020, 1.1446] |
| $e_\omega$ | IG | 1.000 | 2.00 | 0.5725 | [0.2531, 0.9076] |
| $e_\zeta$ | IG | 1.000 | 2.00 | 0.7337 | [0.5082, 0.9978] |
| **South Africa** | | | | | |
| $e_D$ | IG | 1.000 | 2.00 | 16.8841 | [13.4235, 19.8519] |
| $e_S$ | IG | 1.000 | 2.00 | 0.5911 | [0.4591, 0.7783] |
| $e_m$ | IG | 1.000 | 2.00 | 2.5641 | [0.7215, 3.6413] |
| $e_f$ | IG | 1.000 | 2.00 | 1.6158 | [0.6270, 2.9203] |
| $e_\omega$ | IG | 1.000 | 2.00 | 1.6426 | [1.4002, 1.9408] |
| $e_\zeta$ | IG | 1.000 | 2.00 | 1.3664 | [0.5573, 2.2300] |

brings fluctuation in inflation rate in all the sample emerging countries, whereas, the other shocks contribute differently across sample countries.

In Russia, a supply shock brings 99% fluctuations in both inflation and interest rates. The estimates indicate that almost negligible fluctuations occur in inflation and interest rates due to demand, foreign interest rate, and fiscal imbalances shocks in Brazil, Russia, and India. The estimates show that credit spread shock significantly affects inflation in Brazil and South Africa. The results also show that the foreign interest rate, monetary policy, and fiscal imbalances shocks are the major source of fluctuations in output, inflation, and the interest rate in China and South Africa.

**Table 5. Variance decompositions.**

| Variable | Shock | | | | | |
|---|---|---|---|---|---|---|
| | Monetary policy | Demand | Supply | Foreign interest rate | Credit spread | Fiscal Imbalances |
| **Brazil** | | | | | | |
| $y$ | 26.10 | 2.45 | 8.59 | 30.42 | 5.79 | 26.65 |
| $\pi$ | 0.00 | 0.00 | 38.28 | 0.00 | 61.72 | 0.00 |
| $i_d$ | 0.00 | 0.00 | 37.74 | 0.00 | 62.26 | 0.00 |
| $i_{avg}$ | 0.00 | 0.00 | 30.67 | 0.00 | 69.32 | 0.00 |
| **Russia** | | | | | | |
| $y$ | 46.82 | 0.00 | 7.58 | 0.27 | 44.46 | 0.86 |
| $\pi$ | 0.00 | 0.00 | 99.70 | 0.00 | 0.30 | 0.00 |
| $i_d$ | 0.00 | 0.00 | 99.86 | 0.00 | 0.14 | 0.00 |
| $i_{avg}$ | 0.00 | 0.00 | 99.88 | 0.00 | 0.12 | 0.00 |
| **India** | | | | | | |
| $y$ | 2.41 | 0.04 | 39.10 | 3.08 | 0.17 | 55.20 |
| $\pi$ | 0.21 | 0.00 | 89.32 | 0.18 | 9.75 | 0.53 |
| $i_d$ | 0.61 | 0.47 | 21.34 | 0.46 | 76.29 | 0.84 |
| $i_{avg}$ | 0.56 | 0.43 | 19.51 | 0.41 | 78.32 | 0.77 |
| **China** | | | | | | |
| $y$ | 28.03 | 0.02 | 70.09 | 0.79 | 0.52 | 0.54 |
| $\pi$ | 20.46 | 0.29 | 22.94 | 8.74 | 1.09 | 46.47 |
| $i_d$ | 11.94 | 0.75 | 40.28 | 5.76 | 2.35 | 38.92 |
| $i_{avg}$ | 11.79 | 0.74 | 39.77 | 5.69 | 3.52 | 38.49 |
| **South Africa** | | | | | | |
| $y$ | 36.23 | 20.87 | 8.54 | 14.86 | 2.14 | 17.36 |
| $\pi$ | 9.59 | 0.89 | 37.29 | 3.49 | 27.38 | 21.35 |
| $i_d$ | 6.07 | 33.76 | 18.54 | 2.14 | 22.51 | 16.98 |
| $i_{avg}$ | 5.57 | 30.58 | 16.64 | 1.97 | 29.14 | 16.11 |

## 5.3 Impulse response analysis

The dynamic behavior of the model is analyzed by the impulse responses. IRFs illustrate the mechanism through which random innovation changes into the endogenous variables' fluctuations. The temporary shocks to the several structural shocks are presented in S2 Fig (*sub-figures A2.1-A2.5 in S2 Appendix*).

The solid lines of the graph represent the mean values of the posterior distribution, whereas, the thin lines represent 5% and 95% confidence bands.

The estimated IRFs in the sample economies respond to monetary policy shocks in a similar way but with different speed of adjustment. The response of this kind of shock is in accordance with the literature [52, 53, 74]. The IRFs depict that an increase in the interest rate leads to a reduction in the level of output and inflation. The inflation rate will not increase as much with increase in rate of interest due to the sticky prices. Hence, the real rate of interest increases significantly. The demand also decreases, because there is no immediate change in the consumption pattern of the. Due to decrease in demand, firms' production decreases and thus, deflation occurs. The interest rate moves towards the steady state, output returns to steady-state after 15 time periods. Except Russia and China, other sample countries show hump shaped fall in output. The monetary shock persists for less than 5 time periods in Russia and China [62, 63, 66].

A positive demand shock positively affects the demand for goods. When tax rate increases, it positively affects inflation along with the increase in cost of production, and discourages output. Inflation starts decreasing when the production in the economy cuts across its natural level. The central bank intervenes by lowering the interest rate. The nominal interest rate decreases more as compared to inflation. Hence, the real interest rate decreases, increasing the level of output in the long run. On average demand shock persists for 20 time periods. Demand shock does not persist for a longer time period in Russia. These results support the existing literature [71, 74].

The rate of inflation is directly affected by the supply shocks. In Russia and India, supply shocks affect the rate of inflation with a magnitude of more than 89%. The above figure presents the impulse responses of the sample emerging economies. With the increase in the inflation rate, the central bank intervenes by lowering the rate of interest. The interest rate decrease tends to be larger than the decline in the inflation rate. Therefore, the real rate falls and hence, positively impacts on the output level. The fall in in real wage results in the fall of both marginal cost, and inflation rate. On average, supply shocks persist for 15 periods in the sample emerging economies. However, the supply shock does not persist for a longer time period in India. These findings are consistent with the literature [64, 72].

The estimated IRFs to the foreign interest rate shock are presented in S2 Fig for each country. The positive shock increases the return on foreign bonds. Therefore, capital outflow takes place. Households buy more foreign bonds by decreasing their consumption. Initially, this shock does not affect the domestic rate of interest. However, the domestic rate has to be increased for stopping the capital outflow. The inflation rate is not immediately impacted by the increased interest rate. According to the Fisher's rule, the real rate of interest increases, depressing the output level. The firm's production is also affected negatively as a result of a decrease in household demand. In response, deflation occurs in the economy. On average, this shock persists for 20 time periods in the emerging economies. In Russia, this sort of shock recovers after 10 time periods. The results are consistent with the previous studies [13, 71].

Impulse responses to a credit spread shock are presented in S2 Fig. The macroeconomic variables are affected by the contraction of the lending rate. There is a negative impact of increased credit spread on both, the output level, and the inflation rate. The borrowing capacity of firms is affected due to an increase in lending rate. Firms reduce their investments. It also negatively affects the level of output. This sort of shock induces the firms to hire a small number of workers with lowered wages. The consumption of households is also negatively affected by these kinds of shocks. The policy rate must decrease to preserve the initial cost of borrowing. Hence, increased credit spread requires a decrease in the policy rate. On average, this shock persists for 10 time periods in the emerging economies. Again, in Russia, this shock restores before 5 time periods. Our findings support the literature [17, 18].

S2 Fig shows the estimated IRFs of a shock to fiscal imbalances. Fiscal imbalances shock directly affects the rate of inflation, output level and interest rates are also affected. This shock positively affects the inflation rate. The central bank intervenes by decreasing the rate of interest. In response to decreased interest rate, price level increases that discourages the consumers spending, with their reduced purchasing power resulting in the decreased demand. Hence, the output falls in the long run. This shock persists for a longer time period in the case of China and South Africa. These findings are consistent with the existing literature [75]. BRICS economies have experienced budget deficits and debt accumulation in the previous decade. However, Brazil and India showed higher indebtedness and fiscal imbalances from year 2000 to now. The study by Tran [76] reported that these two economies are more vulnerable as compared to countries with low deficits and debts.

## 5.4 Comparison

BRICS economies are highly heterogeneous and it is not possible to conduct a unified MP for the whole union with a single motive. The role of central banks is increasing day by day in these economies. Currently, the MP of BRICS member countries are based on interest rate and on the situation of foreign exchange market. BRICS economies confirm that past central bank actions, particularly adjustments to the base interest rate; significantly influence the overall outcomes of monetary policy. In order to control inflation in BRICS economies, their central banks paid attention to expected inflation, GDP growth and exchange rate fluctuations. Bekareva and Meltenisova [14] recommended considering an exchange rate when implementing a monetary policy in the BRICS countries. The current study considers exchange rate as well as the effect of foreign interest rate in the BRICS economies.

Interest rate is a fee paid by a borrower to a lender reflecting the cost of borrowing and is a valuable tool in monetary policy. It regulates the economic activity and inflation. The data for BRICS countries show that after 2017, interest rates in various countries have remained relatively safe, ranging from 4 to 7%. It is essential to comprehensively consider the current economic situation while formulating the policy. The empirical estimates for BRICS countries show that the largest decrease in interest rates is Brazil, followed by Russia, India, China, and South Africa [77]. There are numeral reasons for the decline in interest rates in these countries. The global financial crisis of 2008 is one of the possible reasons, which led to a sharp decline in economic growth and inflation. In reaction, the central banks in the BRICS countries cut interest rates to stimulate economic growth. In general, the decrease in interest rates in the BRICS economies has had a mixed impact on the economy. It not only boosted economic growth and spending but has also led to a decline in the savings rate and an increase in debt levels.

As discussed earlier, the existence of credit spread departs the monetary authorities to stabilize the domestic prices, whereas, the monetary autonomy helps the authorities to again focus on the stabilization of the prices. The inflation coefficient is reported to be lowest in Russia among other economies. In 1990's money supply was excessively increased in Russia leading to inflation; increased velocity of money directly affects prices. The reason was impoverishment of pensioners, and the loss of the value of lifetime savings. This circle continues till their head of Central Bank was replaced. When Tatyana Paramonova became the head of Central Bank, hyperinflation quickly disappeared. The estimates report the fiscal imbalance coefficient is positive and significant in all the sample economies. The estimates show that fiscal imbalance coefficient is higher in South Africa as compared to other sampled economies. It is considered as one of the main sources of inflation in the emerging economies.

The current study shows that the data is informative regarding the posterior distribution. Except China, other countries estimates show that output coefficient is greater than inflation coefficient indicating more concern for the output. Similarly, interest rate smoothening parameter is higher in India, indicating strong willingness of the central monetary authorities to smooth the movement of interest rate in the economy. As far as the standard deviation of the exogenous processes are concerned, the wider confidence interval indicating insignificant results for the demand shock. However, to demand shock, all the economies show significant estimates except South Africa. Except Russia, all other Countries have strong and higher level of significance to the supply shock. In the same way, except Russia and South Africa the rest of economies show exogenous processes to be significant. The non-systematic part of monetary policy to credit spread shock shows that Brazil and Russia have insignificant standard deviation to the credit spread shock. BRICS economies have experienced budget deficits and debt accumulation in the previous decade. However, Brazil and India showed higher indebtedness and fiscal imbalances from year 2000 to now. The study by Tran

[76] reported that these two economies are more vulnerable as compared to other countries with low deficits and debts.

## 6. Conclusion

In this study, a modified PRF is estimated for the BRICS economies by using Bayesian estimation technique. The estimated parameters appear reasonable and also in accordance with the literature. The posterior estimates confirm that credit spread, fiscal imbalance, and monetary autonomy significantly contribute to the fluctuations in output, inflation rate and the interest rate. The empirical estimates show that fiscal imbalances shock significantly affect output in Brazil, India, and South Africa whereas, inflation and interest rate are significantly affected by fiscal imbalances shock in China and South Africa. The supply shock significantly affects output, inflation, and interest rate in all the sample economies. Except Russia, the credit spread shock substantially impacts inflation in all the sample economies.

Except China, in all the sample BRICS countries, output coefficient is higher than the inflation coefficient. The estimates indicate that the sample emerging economies show more concern for output as compared to inflation. However, the inflation coefficient is also significant in these countries. Impulse responses of the exogenous processes match with the prior expectations. Despite budget deficits, emerging economies do not focus on the tax reforms. The study recommends these economies to focus on tax reforms in order to fill the gap between its revenues and expenditures, and monetary autonomy is also crucially needed for these economies. The persistently huge budget deficit in the emerging economies badly affects the effectiveness of the monetary policy. The effectiveness of fiscal and monetary policy requires coordination among treasury benches and central monetary authority. Like Benchimol, Saadon and Segev [16] and Benchimol and Ivashchenko [6], for future research one can use nonlinear models to address open-economy and market-related variables, which are subject to more nonlinear dynamics. Due to the varied economic structure of the BRICS, these nations cannot successfully implement a unified monetary policy. This research lays the groundwork for predicting future central bank actions in major emerging markets.

## Supporting information

**S1 Fig. Prior and posterior distribution of the structural parameters.**
(ZIP)

**S2 Fig. Impulse responses to different shocks for the emerging countries.**
(ZIP)

**S1 Appendix. Derivation of monetary policy reaction function.**
(DOCX)

**S2 Appendix. Impulse responses to different shock for the emerging countries.**
(DOCX)

**S3 Appendix. Derivation of monetary policy reaction function.**
(DOCX)

## Author Contributions

**Conceptualization:** Farah Waheed.

**Data curation:** Lubna Maroof.

**Methodology:** Farah Waheed.

**Software:** Farah Waheed.

**Supervision:** Abdul Rashid.

**Writing – original draft:** Farah Waheed.

**Writing – review & editing:** Asma Basit.

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
