## [Decision Letter · Decision Letter 0]

22 Mar 2023

PONE-D-23-06648Monetary Policy Reaction Function: A Bayesian AnalysisPLOS ONE

Dear Dr. Farah Waheed

Thank you for submitting your manuscript to PLOS ONE. After careful consideration, we feel that it has merit but does not fully meet PLOS ONE’s publication criteria as it currently stands. Therefore, we invite you to submit a revised version of the manuscript that addresses the points raised during the review process.

We look forward to receiving your revised manuscript.

Kind regards,

Muhammad Kamran Khan, PhD Finance

Academic Editor

PLOS ONE

Journal Requirements:

"NO"

Additional Editor Comments (if provided):

Strenghten your abstract by mentioning your major findings of DSGE model. Suggest recommendation based on SDG Goals and which SDG goal this study cover.

Reviewers' comments:

Reviewer's Responses to Questions

**Comments to the Author**

1. Is the manuscript technically sound, and do the data support the conclusions?

Reviewer #1: Yes

Reviewer #2: Yes

2. Has the statistical analysis been performed appropriately and rigorously? 

Reviewer #1: Yes

Reviewer #2: Yes

3. Have the authors made all data underlying the findings in their manuscript fully available?

Reviewer #1: Yes

Reviewer #2: Yes

4. Is the manuscript presented in an intelligible fashion and written in standard English?

Reviewer #1: Yes

Reviewer #2: Yes

5. Review Comments to the Author

Reviewer #1: Dear authors,

I'd like to congratulate you and your team on your excellent research work in your paper submitted for publication in this prestigious journal. The topic is very interesting, and I enjoyed it. I would like to thank you for your efforts in presenting your research work in such a professional manner. However, before your work is recommended or accepted, a few comments must be included/ addressed to improve the quality of your work as well as for future publication in this reputable journal. I have the following observations, questions, and comments that may help to improve your work. The authors must modify the following points in great detail.

1. In the abstract, please include 2-3 special quantitative achievements from the findings of this study in the context of the environment by combining the research objectives and problems. Please limit your abstract to 250 words. Check spellings for many words that are misspelt or written in haste.

2. The introduction section needs a few more sentences to strengthen the article, and please include the research problem, objective, and novelty in the last paragraph of the Introduction section.

3. Include a few more sentences at the beginning of the introduction explaining your paper's contribution to the environment, climate change impact, and sustainability, as well as your attempts to deal with or present solutions to a specific problem/s and your unique contribution with this research paper.

4. Please also present the methodology section in a concise graphical format.

5. The literature review section is very weak; please revise it.

I found that the literature section is a little weak, shift your study a little more towards environment friendly and sustainability, therefore it requires more studies to be reviewed therefore I suggest you to include the following work:

https://doi.org/10.1007/s11356-023-25574-9

https://doi.org/10.1007/s11356-021-15421-0

https://doi.org/10.1007/s11356-021-14745-1

https://doi.org/10.1007/s11356-021-13441-4

https://doi.org/10.1007/s10668-021-01418-9

I think above all studies will make this study more relevant in bridging the gap with literature.

Looking forward for your revised submission.

Reviewer #2: The manuscripts is well written and is in line with the journal requirement; both research and publication ethics. The language too is standard. Tables, equations and figures are well presented including those are the appendices. The authors make the following adjustments:

1. The DSGE in the Keywords should be defined and mentioned in the Abstract.

2. "Monetary policy rules" in the Keywords is also not mentioned in the Abstract, although it has been mentioned in the Introduction.

6. PLOS authors have the option to publish the peer review history of their article (what does this mean?). If published, this will include your full peer review and any attached files.

Reviewer #1: **Yes: **Vishal

Reviewer #2: No

While revising your submission, please upload your figure files to the Preflight Analysis and Conversion Engine (PACE) digital diagnostic tool, https://pacev2.apexcovantage.com/. PACE helps ensure that figures meet PLOS requirements. To use PACE, you must first register as a user. Registration is free. Then, login and navigate to the UPLOAD tab, where you will find detailed instructions on how to use the tool. If you encounter any issues or have any questions when using PACE, please email PLOS at figures@plos.org. Please note that Supporting Information files do not need this step.<quillbot-extension-portal></quillbot-extension-portal>

---

## [Author Response · Author response to Decision Letter 0]

15 Aug 2023

the suggested changes have been incorporated in the revised manuscript. Table 1 (page 10 of manuscript) details the parameters used for data collection. Changes as per instruction done on page 24 of manuscript.

---

## [Decision Letter · Decision Letter 1]

19 Dec 2023

PONE-D-23-06648R1Monetary Policy Reaction Function: A Bayesian AnalysisPLOS ONE

Dear Dr. Waheed,

Thank you for submitting your manuscript to PLOS ONE. After careful consideration, we feel that it has merit but does not fully meet PLOS ONE’s publication criteria as it currently stands. Therefore, we invite you to submit a revised version of the manuscript that addresses the points raised during the review process.

Specifically, please try to address the comments by the referee who is an expert in the field.

We look forward to receiving your revised manuscript.

Kind regards,

Petre Caraiani

Academic Editor

PLOS ONE

Reviewers' comments:

Reviewer's Responses to Questions

**Comments to the Author**

1. If the authors have adequately addressed your comments raised in a previous round of review and you feel that this manuscript is now acceptable for publication, you may indicate that here to bypass the “Comments to the Author” section, enter your conflict of interest statement in the “Confidential to Editor” section, and submit your "Accept" recommendation.

Reviewer #3: (No Response)

2. Is the manuscript technically sound, and do the data support the conclusions?

Reviewer #3: Partly

3. Has the statistical analysis been performed appropriately and rigorously? 

Reviewer #3: No

4. Have the authors made all data underlying the findings in their manuscript fully available?

Reviewer #3: No

5. Is the manuscript presented in an intelligible fashion and written in standard English?

Reviewer #3: Yes

6. Review Comments to the Author

Reviewer #3: See the attached report.

Please address all the comments from this report and invest time in enriching your paper according to those comments.

7. PLOS authors have the option to publish the peer review history of their article (what does this mean?). If published, this will include your full peer review and any attached files.

Reviewer #3: No

---

## [Author Response · Author response to Decision Letter 1]

2 Feb 2024

Editorial revisions 

Sr. No Comment Decision /section Reference 

1 The paper is difficult to read and does not expose a clear argumentative path to the research question, which is to estimate MPRFs for specific countries. To address this issue, the authors should clarify the set of countries analyzed and the reason behind this choice. I suggest the authors focus on the BRICS and compare their results with Pakistan (and maybe other countries that recently entered the BRICS in another section). Incorporated in Introduction Thank you for useful suggestions. 

The paper structure has been improved by incorporating the recommendation of the reviewer. 

Relevant sections are highlighted on page 2 and page 4. 

2 a) The calibration of priors is not appropriate for a comparative study. All the parameters should be calibrated to the same levels for all countries. Same for density functions. Calibrate the estimations for each country the same to let the data speak. 

b) The paper briefly mentions identification issues. This aspect requires more attention; some tests should be done, and a thorough discussion should appear, particularly regarding the potential sensitivity of results to the fixed parameters. Identification tests which all are integrated into the Dynare toolbox, must be calculated, and results displayed or at least presented in the discussion about identification Incorporated a) Suggested changes are incorporated on Page 11, 12 and 13 (highlighted). 

b) Suggested changes are incorporated on Page 10. 

Furthermore, the analysis is conducted in MATLAB using Dynare tool. 

3 a) Compare the obtained rules with the historical central bank rate decisions or market interest rates to assess the estimates concerning the MPRF and how close they are to effective rates (reality). This comparison should also include one or two standard rules appropriate to BRICS, and computing the RMSE for all of these rules should demonstrate that the authors’ results overperform the classical ones for BRICS in some ways. 

b) at least compute some loss functions to show that their MPRFs are appropriate according to the respective central bank objectives. Incorporated 

a) the justification of taking the union of BRICS for analysis is highlighted on page 2

b) Already done in the deriving the MPRF and duly referred to Waheed and Rashid (2021). Highlighted on page 4. 

4 Some discussion on critical points should appear in the paper. First, the fact that the considered economies are open or small open, thus being influenced by the foreign economy potentially in a nonlinear way. As transforming the current model into a fully nonlinear model could be a cumbersome task, policy discussions to provide the reader with a perspective on the nonlinearities that may occur between the domestic and foreign countries. Second, and more generally, a discussion on policy implications derived from these analyses and results is also recommended. Incorporated The suggestion is considered and write up in improved in general. 

5 References 

The below reference appears in the bibliography: Korinek, A., & Sandri, D. (2016). Capital controls or macroprudential regulation? Journal of International Economics, 99(S), S27-S42. However, this reference is not cited in the text. I suggest the authors check if each reference entry is cited in the text and if each citation appears in the bibliography. Incorporated Mentioned reference is cross checked and removed. All the in text references are bibliography is verified. 

6 Title 

The current title is too general to attract relevant readers. I suggest the authors include “for the BRICS” in their new title. Incorporated The title is revised to include “BRICS”, highlighted on page 1. 

7 Conclusion 

The concluding remarks should provide clear avenues for future research, building on the insights gained from the current paper. Incorporated 

The clear avenue for future research is improved, details added and additions highlighted on page 22.

---

## [Decision Letter · Decision Letter 2]

8 Feb 2024

PONE-D-23-06648R2Monetary Policy Reaction Function: A Bayesian Analysis for the BRICSPLOS ONE

Dear Dr. Waheed,

Thank you for submitting your manuscript to PLOS ONE. After careful consideration, we feel that it has merit but does not fully meet PLOS ONE’s publication criteria as it currently stands. Therefore, we invite you to submit a revised version of the manuscript that addresses the points raised during the review process.

Specifically, I agree with the comments by the referee: *"The authors have to comply with ALL the comments. They have to address those comments seriously; PLOS One is a reputable journal, and addressing them seriously is requested."*

We look forward to receiving your revised manuscript.

Kind regards,

Petre Caraiani

Academic Editor

PLOS ONE

Reviewers' comments:

Reviewer's Responses to Questions

**Comments to the Author**

1. If the authors have adequately addressed your comments raised in a previous round of review and you feel that this manuscript is now acceptable for publication, you may indicate that here to bypass the “Comments to the Author” section, enter your conflict of interest statement in the “Confidential to Editor” section, and submit your "Accept" recommendation.

Reviewer #3: (No Response)

2. Is the manuscript technically sound, and do the data support the conclusions?

Reviewer #3: No

3. Has the statistical analysis been performed appropriately and rigorously? 

Reviewer #3: No

4. Have the authors made all data underlying the findings in their manuscript fully available?

Reviewer #3: No

5. Is the manuscript presented in an intelligible fashion and written in standard English?

Reviewer #3: No

6. Review Comments to the Author

Reviewer #3: The authors have to comply with ALL the comments. They have to address those comments seriously; PLOS One is a reputable journal, and addressing them seriously is requested.

7. PLOS authors have the option to publish the peer review history of their article (what does this mean?). If published, this will include your full peer review and any attached files.

Reviewer #3: No

---

## [Author Response · Author response to Decision Letter 2]

23 Mar 2024

Sr. No Comment Decision /section Reference 

1 The calibration of priors is still not appropriate for a comparative study. All the parameters should be calibrated to the same levels for all countries. Same for density functions. Calibrate the estimations for each country the same to let the data speak. Table 2 shows that this comment was ignored. The calibration of priors refers to both the prior mean and prior SD. 

Suggested changes are incorporated on Page 12, 13, 16 and 17 (highlighted). 

2 The paper briefly mentions identification issues. This aspect requires more attention; some tests should be done, and a thorough discussion should appear, particularly regarding the potential sensitivity of results to the fixed parameters. Identification tests of Iskrev (2010), Komunjer and Ng (2011), Qu and Tkachenko (2012), and Ivashchenko and Mutschler (2020), which all are integrated into the Dynare toolbox, must be calculated, and results displayed or at least presented in the discussion about identification. This comment was not addressed. 

Thank you for useful suggestions. Suggested changes are incorporated on Page 11 and 12 (highlighted). Furthermore, the analysis is conducted in MATLAB using Dynare tool as mentioned at page 11. Mod files of all sample countries are available on request.

3 Compare the obtained rules with the historical central bank rate decisions or market interest rates to assess the estimates concerning the MPRF and how close they are to effective rates (reality). This comparison should also include one or two standard rules appropriate to BRICS, and computing the RMSE for all of these rules should demonstrate that the authors’ results overperform the classical ones for BRICS in some ways. 

Suggested changes are incorporated on Page 22 (highlighted). 

4 The authors should cite and follow the methodology of Benchimol and Fourçans (2019) and at least compute some loss functions to show that their MPRFs are appropriate according to the respective central bank objectives. They could also quickly discuss optimal policy and Benchimol and Bounader (2023). This comment was also ignored. The authors have to compute several loss functions (at least several weights) and build these loss functions according to the “ideal” central bank objective and/or according to each central bank objective. Also, their citation of Benchimol and Bounader (2023) is inappropriate; the latter do not deal with forecasting questions 

Already done in the deriving the MPRF and duly referred to Waheed and Rashid (2021). Highlighted on page 4. Complete derivation is Presented in Appendix now. Suggested changes are incorporated at page 5 and 11.

5 Some discussion on critical points should appear in the paper. First, the fact that the considered economies are open or small open, thus being influenced by the foreign economy potentially in a nonlinear way. As transforming the current model into a fully nonlinear model could be a cumbersome task, I suggest the authors discuss Benchimol and Ivashchenko (2021) in their results and policy discussions to provide the reader with a perspective on the nonlinearities that may occur between the domestic and foreign countries. Second, and more generally, a discussion on policy implications derived from these analyses and results is also recommended. Third, the authors should discuss the interaction of the estimated monetary policy functions with uncertainty regarding the financial market by citing and relating Benchimol, Saadon and Segev (2023) to their results. Comments completely ignored. 

The suggestion is considered and incorporated in this research study at page 3, 6 and 23(highlighted).

---

## [Decision Letter · Decision Letter 3]

3 Apr 2024

PONE-D-23-06648R3Monetary Policy Reaction Function: A Bayesian Analysis for the BRICSPLOS ONE

Dear Dr. Waheed,

Thank you for submitting your manuscript to PLOS ONE. After careful consideration, we feel that it has merit but does not fully meet PLOS ONE’s publication criteria as it currently stands. Therefore, we invite you to submit a revised version of the manuscript that addresses the points raised during the review process.

We look forward to receiving your revised manuscript.

Kind regards,

Petre Caraiani

Academic Editor

PLOS ONE

Reviewers' comments:

Reviewer's Responses to Questions

**Comments to the Author**

1. If the authors have adequately addressed your comments raised in a previous round of review and you feel that this manuscript is now acceptable for publication, you may indicate that here to bypass the “Comments to the Author” section, enter your conflict of interest statement in the “Confidential to Editor” section, and submit your "Accept" recommendation.

Reviewer #3: (No Response)

2. Is the manuscript technically sound, and do the data support the conclusions?

Reviewer #3: Partly

3. Has the statistical analysis been performed appropriately and rigorously? 

Reviewer #3: Yes

4. Have the authors made all data underlying the findings in their manuscript fully available?

Reviewer #3: No

5. Is the manuscript presented in an intelligible fashion and written in standard English?

Reviewer #3: No

6. Review Comments to the Author

Reviewer #3: See the report attached.

Please address all the comments in the report (in red) seriously.

Read the suggested references to better use them for your paper (discussions).

7. PLOS authors have the option to publish the peer review history of their article (what does this mean?). If published, this will include your full peer review and any attached files.

Reviewer #3: No

---

## [Author Response · Author response to Decision Letter 3]

20 May 2024

1 Policy Research Question

 The apparent policy research question is not strong enough to be published. Indeed, running estimations and getting parameters of monetary policy rules is not enough for academic or policy research. The authors should add two critical sections to this paper.

a) Compare the obtained rules with the historical central bank rate decisions or market interest rates to assess the estimates concerning the MPRF and how close they are to effective rates (reality). This comparison should also include one or two standard rules appropriate to BRICS, and computing the RMSE for all of these rules should demonstrate that the authors’ results overperform the classical ones for BRICS in some ways.

b) The authors should cite and follow the methodology of Benchimol and Fourçans (2019) and at least compute some loss functions to show that their MPRFs are appropriate according to the respective central bank objectives. Incorporated 

Thank you for the useful suggestions. 

Relevant sections are highlighted on page 23

Suggested changes are incorporated on Page 2

2 Discussion 

Some discussion on critical points should appear in the paper. First, the fact that the considered economies are open or small open, thus being influenced by the foreign economy potentially in a nonlinear way. As transforming the current model into a fully nonlinear model could be a cumbersome task, I suggest the authors discuss Benchimol and Ivashchenko (2021) in their results and policy discussions to provide the reader with a perspective on the nonlinearities that may occur between the domestic and foreign countries. Second, and more generally, a discussion on policy implications derived from these analyses and results is also recommended. Third, the authors should discuss the interaction of the estimated monetary policy functions with uncertainty regarding the financial market by citing and relating Benchimol, Saadon and Segev (2023) to their results. 

Incorporated Suggested changes are incorporated on Page 6 and 25 (highlighted). 

3 The authors have to check if each paper cited in the text is appropriately referenced in the bibliography, and reciprocally. This is currently not the case 

Incorporated 

All the in text references and bibliography is cross checked.

---

## [Decision Letter · Decision Letter 4]

4 Jun 2024

PONE-D-23-06648R4Monetary Policy Reaction Function: A Bayesian Analysis for the BRICSPLOS ONE

Dear Dr. Waheed,

Thank you for submitting your manuscript to PLOS ONE. After careful consideration, we feel that it has merit but does not fully meet PLOS ONE’s publication criteria as it currently stands. Therefore, we invite you to submit a revised version of the manuscript that addresses the points raised during the review process. 

We look forward to receiving your revised manuscript.

Kind regards,

Petre Caraiani

Academic Editor

PLOS ONE

Reviewers' comments:

Reviewer's Responses to Questions

**Comments to the Author**

1. If the authors have adequately addressed your comments raised in a previous round of review and you feel that this manuscript is now acceptable for publication, you may indicate that here to bypass the “Comments to the Author” section, enter your conflict of interest statement in the “Confidential to Editor” section, and submit your "Accept" recommendation.

Reviewer #3: (No Response)

2. Is the manuscript technically sound, and do the data support the conclusions?

Reviewer #3: Partly

3. Has the statistical analysis been performed appropriately and rigorously? 

Reviewer #3: Yes

4. Have the authors made all data underlying the findings in their manuscript fully available?

Reviewer #3: No

5. Is the manuscript presented in an intelligible fashion and written in standard English?

Reviewer #3: Yes

6. Review Comments to the Author

Reviewer #3: See the attached referee report (PDF file).

Some comments were addressed but some others need careful attention.

7. PLOS authors have the option to publish the peer review history of their article (what does this mean?). If published, this will include your full peer review and any attached files.

Reviewer #3: No

---

## [Author Response · Author response to Decision Letter 4]

1 Jul 2024

Sr. No Comment Decision /section Reference 

1 On page 12, the authors do mention that the fit of the model to the data will be analyzed (marginal likelihood), but the fit of each estimation is not assessed. A table comparing the log marginal data densities across all estimations is needed. 

Incorporated Thank you for the useful suggestions. These changes are highlighted on page 13, 14, 15 and 16.

2 This adds to my comment that still stays unaddressed. The authors have to compare the realized interest rate, i.e., the interest rate recommended by the estimated model, with the realized interest rate (effective). This can be done through simple RMSE, but any other methodology comparing the global fitting of the model AND the specific fitting of the interest rate is necessary to confirm the robustness and relevance of the estimations for the specific research question. For the moment, this has not been done 

 Incorporated Suggested changes are incorporated on Page 19. 

3 On a more general perspective, there is not enough discussion of the results and comparisons with the existing literature. This comment stays up to the fifth round (!), so I conclude the authors do not want to invest efforts in this task, while this would bring value to the final reader and the overall readability of the paper and the originality of the obtained results. 

 Incorporated Suggested changes are incorporated on Page 19, 23 and 24.

4. On page 2, the sentence "In the recent era, some policy rules have gained extensive attention for designing a visible and an effective monetary policy" should cite this fresh paper: 10.1016/j.jmacro.2024.103604 (and the reference has to appear in the bibliography, see below comment) 

 Incorporated Suggested changes are incorporated on Page 2

5 Again, the authors have to check if each paper cited in the text is appropriately referenced in the bibliography, and reciprocally. For instance, Benchimol and Fourçans (2019) does not appear in the bibliography while it is cited in the main text. Thus carefully checking all the references appearing in the text and the bibliography is necessary. 

 Incorporated All the references are cross checked.

---

## [Decision Letter · Decision Letter 5]

5 Jul 2024

Monetary Policy Reaction Function: A Bayesian Analysis for the BRICS

PONE-D-23-06648R5

Dear Dr. Waheed,

We’re pleased to inform you that your manuscript has been judged scientifically suitable for publication and will be formally accepted for publication once it meets all outstanding technical requirements.

Kind regards,

Petre Caraiani

Academic Editor

PLOS ONE

Additional Editor Comments (optional):

Reviewers' comments:

Reviewer's Responses to Questions

**Comments to the Author**

1. If the authors have adequately addressed your comments raised in a previous round of review and you feel that this manuscript is now acceptable for publication, you may indicate that here to bypass the “Comments to the Author” section, enter your conflict of interest statement in the “Confidential to Editor” section, and submit your "Accept" recommendation.

Reviewer #3: All comments have been addressed

2. Is the manuscript technically sound, and do the data support the conclusions?

Reviewer #3: Yes

3. Has the statistical analysis been performed appropriately and rigorously? 

Reviewer #3: Yes

4. Have the authors made all data underlying the findings in their manuscript fully available?

Reviewer #3: Yes

5. Is the manuscript presented in an intelligible fashion and written in standard English?

Reviewer #3: Yes

6. Review Comments to the Author

Reviewer #3: (No Response)

7. PLOS authors have the option to publish the peer review history of their article (what does this mean?). If published, this will include your full peer review and any attached files.

Reviewer #3: No

---

## [Editor Report · Acceptance letter]

15 Jul 2024

PONE-D-23-06648R5 

PLOS ONE

Dear Dr. Waheed, 

I'm pleased to inform you that your manuscript has been deemed suitable for publication in PLOS ONE. Congratulations! Your manuscript is now being handed over to our production team.

Kind regards, 

on behalf of

Dr. Petre Caraiani 

Academic Editor

PLOS ONE